# Motivating factors for physical activity participation among individuals with chronic obstructive pulmonary disease: A qualitative study applying the motivation, opportunity, and ability model

**Yuanyu Liao** [1,2], **Jiaohua Yu** [1]*, **Yuxin Zhan**[3], **Yunfang Liu**[4], **Yaoling Zhou**[2,5], **Huan Wang**[4], **Xinghong Liu**[6], **Weiwei Wang**[2], **Yu Ma**[2], **Fenfen Lan**[2]

1 Nursing Department, Tongji Medical College Affiliated Union Hospital, Huazhong University of Science and Technology, Wuhan, Hubei Province, China, 2 School of Nursing, Tongji Medical College, Huazhong University of Science and Technology, Wuhan, Hubei Province, China, 3 Department of Neurosurgery, Tongji Medical College Affiliated Union Hospital, Huazhong University of Science and Technology, Wuhan, Hubei Province, China, 4 Department of Thoracic Surgery, Tongji Medical College Affiliated Union Hospital, Huazhong University of Science and Technology, Wuhan, Hubei Province, China, 5 School of Nursing, Huanggang Polytechnic College, Huanggang, Hubei Province, China, 6 Department of Cardiovascular Surgery, Tongji Medical College Affiliated Union Hospital, Huazhong University of Science and Technology, Wuhan, Hubei Province, China

* yujiaohua2008@126.com

## Abstract

### Objective

The study aims to explore the driving forces behind physical activity engagement among patients with chronic obstructive pulmonary disease, focusing on motivation, opportunity, and capability.

### Design

A phenomenological qualitative study applied the motivation, opportunity, and capability model, conducted in two respiratory units of a Chinese university hospital.

### Methods

Participants, selected by age, gender, and illness duration, included inpatients during the interview sessions and those recently discharged within six months. One-on-one semi-structured interviews were recorded, transcribed, and analyzed by the Colaizzi seven-step method.

### Results

Seventeen participants diagnosed with chronic obstructive pulmonary disease for over one year aged between 66 (range: 42–96) participated. Three major themes were identified: Inspiring participation motivation—transitioning from recognizing significance to habit

**Data Availability Statement:** All relevant data are within the paper and its Supporting Information files.

**Funding:** The research received funding from two sources: Sponsor 1, the School of Nursing at Tongji Medical College, Huazhong University of Science and Technology, via the 2023 Independent Innovation Fund (ZZCX2023X003); and Sponsor 2, the Hubei Province Finance Department's 2024 Research Program (08.01.24011). Both grants were awarded to the corresponding author, Jiaohua Yu. Sponsor 1 subsidized the study design, data collection, and analysis; whereas Sponsor 2 funded the manuscript publication and preparation. Detailed information about Sponsor 1 is accessible at http://huli.tjmu.edu.cn/, and Sponsor 2's details are available at https://czt.hubei.gov.cn/.

**Competing interests:** The authors declare that they have no known competing financial interests or personal relationships that could have appeared to influence the work reported in this paper.

formation; Offering participation opportunities—reiterating demand for personalized strategies and ideal environmental settings; Enhancing participation capability—addressing strategies for overcoming fears, setting goals, ensuring safety, and adjusting activity levels.

## Conclusions

This research underscores the vital role of inspiring participation motivation, offering opportunities, and enhancing the capability for participation in effective engagement. Advocating increased attention from healthcare departments, fostering interdisciplinary collaboration, improving activity guidance and counseling effectiveness, and considering individual preferences can significantly benefit those patients with chronic obstructive pulmonary disease who hesitate or are unable to participate in physical activities, thereby increasing the dose of non-leisure time physical activity.

## Introduction

Chronic Obstructive Pulmonary Disease (COPD) is presently one of the top three causes of global mortality, with 90% of fatalities occurring in middle- and low-income countries [1, 2]. Statistics indicate an overall COPD prevalence of 8.6% in China, soaring to 13.7% among individuals aged 40 and above. Nearly 100 million individuals are affected, surpassing 27% in the 60 and above age group, with 5 to 11 million people becoming disabled due to COPD annually [3]. Simultaneously, COPD is among the four major non-communicable diseases, subjecting many patients to prolonged suffering and premature mortality due to the disease or its complications. United Nations Sustainable Development Goal 3.4 aims, by 2030, to reduce premature mortality, including COPD, by one-third from 2015 levels and enhance the mental well-being and quality of life of these patients [4]. Given the complex pathophysiology of COPD, non-pharmacological interventions, including Physical Activity (PA), significantly impact both the quality of life and prognosis, demonstrating substantial socioeconomic benefits [5].

PA refers to any bodily activity causing energy expenditure through skeletal muscle contractions, augmenting energy consumption beyond basal metabolism [6]. Regular PA proves advantageous in reducing mortality risks and extending life expectancy among COPD patients [7]. BMJ's Best Practice recommends PA for all COPD patients [8]. However, the 2023 revised edition of the Global Initiative for Chronic Obstructive Lung Disease (GOLD) explicitly outlines the ongoing challenges faced by COPD patients in promoting and sustaining PA [9]. Recent cross-sectional study findings in 734 COPD patients across seven Latin American countries revealed 37.9% with low PA levels, averaging 36.1 hours of seated time weekly, 23% engaged in moderate PA, and 39.1% in high PA [10]. Furthermore, in a three-month cohort study by Dragnich et al., baseline mean scores on the Physical Activity Scale for the Elderly (PASE) among 35 COPD patients were 100.2±48.4, declining to 82.0±53.7 after a three-month follow-up, with daily mean steps reducing from 2,483±2,141 at baseline to 1,897±1,500 after three months [11]. The positive effects dissipate within two weeks if activity levels significantly decrease and within 2 to 8 months if physical exercise is not resumed [12], highlighting the importance of maintaining PA. However, only a few studies [13, 14] have recently reported on the maintenance or increase in PA among COPD patients following pulmonary rehabilitation.

Recent studies [15–18] identified factors influencing PA participation among COPD patients, linking established factors to demographic (e.g., gender, marital status), disease-

related (e.g., comorbidities), psychological (e.g., motivation, intent, exercise perception, self-efficacy, feedback and improvement capacity, habit formation, fear of breathlessness, anxiety, depression, kinesiophobia), and social factors (relationship with healthcare providers, peer interactions, social support, engagement opportunities). While factors influencing COPD patient PA have been previously identified, they fall short of explaining the dynamics behind complex PA engagement behaviors. Gaining deeper insights into the driving forces behind PA engagement in COPD patients is imperative to foster sustained PA and improve prognosis.

Various theories, including social cognitive theory, humanistic framework, socioecological framework, dual process models, and cross-theoretical models, facilitate understanding and modification of PA, each with specific strengths and limitations [19]. A study [20] employed social cognitive theory to evaluate PA participation barriers and facilitators in COPD, yet it encountered problems where individual positive intentions did not correspond with actual behavior [21]. Additionally, researchers [22] used the Physical Activity-Related Health Competence (PAHCO) model to predict PA levels at 6 weeks and 6 months post-pulmonary rehabilitation in 350 COPD patients. Although effective with physically impacted and non-communicable diseases, this model may overlook subjective elements, such as disease-specific fears. Another study [23], drawing on the health belief model and self-determination theory, probed COPD patients' non-participation in sports activities. Both theories are micro-theories, emphasizing critical construct relationships. Alternatively, utilizing the Capability, Opportunity, Motivation–Behavior (COM-B) model, Midthun and colleagues [24] reviewed patients' home pulmonary rehabilitation programs experience. This model acknowledges the role of individual, group, and environmental factors but is unbiased toward any specific perspective.

In contrast, the Motivation-Opportunity-Ability (MOA) model, with a robust evidence-based structure and operational pathways for understanding behavior change, has been utilized to explain the generation of various complex behaviors [25, 26]. This model provides a comprehensive analytical framework for elucidating the driving forces behind individual information behaviors from three perspectives: subjective likelihood, objective feasibility, and subjective-to-objective cognitive likelihood [27]. It emphasizes exploring the direct impact of motivational factors on behavior and the moderating effects of opportunity and ability factors. Despite the potential value of applying the MOA model in health guidance, its use in research, particularly in explaining the dynamics of COPD patient PA participation, remains limited. Therefore, this study aims to explore, based on the MOA model, the motivation, opportunity, and ability of COPD patients to participate in physical activities, laying the groundwork for developing more patient-friendly interventions promoting and sustaining PA.

## Materials and methods

### Study design

**Participant selection and recruitment.** This study, guided by phenomenological research principles, employed a purposive sampling strategy across two distinct wards within the Respiratory Department of Tongji Medical College Affiliated Union Hospital, Wuhan, Hubei Province, China. It involved face-to-face interviews with hospitalized patients suffering from COPD as well as recruiting patients discharged within the past six months through telephone follow-ups. Participants were required to meet the 2023 Global Initiative for Chronic Obstructive Lung Disease diagnostic criteria for COPD for at least one year, be aged 18 years or older, possess clear consciousness, articulate their views coherently, sign informed consent forms, and voluntarily participate in the study. Exclusion criteria encompassed individuals with impaired organ function (such as heart, liver, kidney, etc.) or other life-threatening conditions,

as well as those unable to communicate due to severe cognitive impairments, communication barriers, or mental illness.

Semi-structured interviews were conducted with individuals of varying ages, genders, and disease durations diagnosed with COPD to capture diverse participant perspectives. Nineteen individuals were recruited, with two excluded from the study. One participant, despite reading and signing the consent form, exhibited evident confusion regarding the research objectives at the commencement of the recorded interview. The other was excluded after confirming during the interview that they had not been diagnosed for over a year. Consequently, between October 26, 2023, and December 15, 2023, data from interviews with 17 individuals (13 males, 4 females) were collected (refer to Table 1 for details). The research took place in locations preferred by the participants (such as hospital rooms or corners of offices), except for a 96-year-old participant who, due to hearing impairment, required caregiver assistance and participated

**Table 1. Patient characteristics (n = 17).**

| Characteristics | Values |
|---|---|
| Age, yr, median (range) | 66(42–96) |
| Sex, n (%) | |
| Male | 13(76.5) |
| Female | 4(23.5) |
| Context, n (%) | |
| Inpatient participants | 12(70.6) |
| Community residents discharged in the last six months | 5(29.4) |
| Educational level, n (%) | |
| Primary school | 4(23.5) |
| Junior high school | 6(35.3) |
| Senior high school | 5(29.4) |
| Junior college or above | 2(11.8) |
| Marital status, n (%) | |
| Married | 15(88.2) |
| Widowed | 2(11.8) |
| Occupational status[a], n (%) | |
| Office workers | 2(11.8) |
| Agricultural producers | 1(5.9) |
| Other employees | 1(5.9) |
| Retirees | 13(76.5) |
| Residential area, n (%) | |
| Cities | 10(58.8) |
| Countryside | 7(41.2) |
| Body mass index, kg/m$^2$, mean±SD | 20.77±3.7 |
| Current smokers, n (%) | 3(17.6) |
| No. of years smoking, mean±SD | 17.59±18.81 |
| No. of packs/d, mean±SD | 0.68±0.82 |
| No. of comorbidities[b], n, median (range) | 1(0–3) |
| Duration of COPD diagnosis, yr, mean±SD | 5.82±5.08 |
| No. of hospitalizations due to deterioration of illness last year, mean±SD | 2.59±1.5 |

[a]Refer to the People's Republic of China (PRC) Occupational Classification Ceremony (2022), and classify the occupations most frequently reported by patients. [b]Most commonly reported comorbidities included diabetes, hypertension, coronary heart disease, tuberculosis, asthma, and atrial fibrillation.

in the interview by their bedside. All other interviews were conducted in the absence of additional individuals. To ensure the quality of our study, we adhered to the Consolidated Criteria for Reporting Qualitative Research (COREQ) [28] (data in S1 Appendix).

**Data collection.**   The interview guide was developed based on the MOA model, integrating expert opinions on the foundation of a literature review. Three participants underwent one-on-one face-to-face pre-interviews, and the research team iteratively discussed and revised the guide (data in S2 Appendix). All interviews were recorded. Throughout the interview process, researchers attentively listened, capturing participants' non-verbal expressions such as body language and facial cues. Various interview techniques, including probing, follow-up questions, and repetition, were employed to elicit authentic responses from participants.

A professional transcriber meticulously transcribed the interviews verbatim within 24 hours of completion, ensuring the complete removal of any identifiable information. Transcripts were returned to participants for corrections. Data saturation was considered achieved when additional interviews failed to yield new themes [29, 30]. Interviews with patients suffering from COPD ranged from 25 to 48 minutes, with an average duration of 32.7 minutes.

**Research team and reflexivity.**   The interview was conducted by a female master's student in nursing (YL) who was trained in qualitative research methods and interview techniques. She approached the research topic impartially, without any preconceived assumptions. Before the interviews, she completed a 2-week clinical nursing internship in the research unit, familiarizing herself with the nursing manager and nurses. She did not know any of the participants before this. After gathering information from the responsible nurses about potential participants' conditions, personalities, financial situations, and social relationships, she introduced herself, the purpose, and the significance of the study to the participants either face-to-face by the bedside or over the phone, ensuring that participants understood the significance of their involvement in the study. Additionally, appreciation for the participants was expressed by observing their PA behaviors during hospitalization, indicating interest in their perspectives and experiences. At the beginning of the interviews, she provided participants with an overview of the study background, objectives, methods, potential risks, risk mitigation measures, and contact information for the research team, ensuring that participants understood how their information would be used, which helped to ensure the efficiency and effectiveness of the interviews.

**Data analysis.**   The Colaizzi method [31] was employed to analyze the data, with the following key steps: (1) Familiarization with interview content: Researchers (YL and YZ) thoroughly and repeatedly read the collected texts, combining on-site notes or written records for a comprehensive assessment; (2) Identification of meaningful statements: Researchers underlined keywords and phrases in the interview content relevant to the research questions; (3) Construction of meaningful units: Researchers coded important recurring statements, aiming to uphold objective information and, when possible, set aside initial assumptions; (4) Clustering of themes: Researchers extrapolated all meaningful units into preliminary themes; (5) Detailed description: Researchers adapted statements based on original data to identify and describe preliminary themes; (6) Generation of a basic structure: YL and YZ individually compared similar preliminary themes, distilled similar ideas, constructed meaningful phrases, and revised them after preliminary discussions with JY; (7) Validation of the developed structure: Two additional researchers (WW and YM) sought feedback from participants to validate the accuracy and credibility of the generated themes. YL and YZ used Microsoft Excel 2021 for data coding and management. After a second round of discussions involving all authors, the final theme was formulated. Subsequently, YL and YZ independently extracted key quotations supporting these findings.

**Ethical considerations.** The present study obtained approval from the Medical Ethics Committee of Tongji Medical College, Huazhong University of Science and Technology, under reference number [2023] Ethics Review Letter (S133). Written informed consent was obtained from all participants.

## Results

From the data, three major themes were identified, including 1) Inspiring Participation Motivation—transitioning from recognizing significance to habit formation; 2) Offering Participation Opportunities—reiterating demand for personalized strategies and ideal environmental settings; 3) Enhancing Participation Capability—addressing strategies for overcoming fears, setting goals, ensuring safety, and adjusting activity levels.

Inspiring Participation Motivation—transitioning from recognizing significance to habit formation

### Understanding significance

The majority of inpatient participants elaborated on the concept and significance of PA, discussing their increasing focus on the benefits of maintaining involvement in such activities. "PA encompasses work, exercise, and daily life, representing a broad concept." (Inpatient participant #5, male, 66 yrs) ". . .I can only rely on exercise to improve resilience. . ." (Inpatient participant #8, male, 42 yrs) However, there was considerable variation in understanding PA among some patients. "I used to work in construction when I was young, bricklaying and concrete work were all physical activities. . . Now I farm at home. . . Besides work, I don't have the habit of exercising, nor do I understand how it could help my condition." (Inpatient participant #3, male, 60 yrs)

### Desire to improve symptoms

Participants expressed a desire to improve respiratory difficulties and enhance overall fitness through the benefits of PA. Some community participants, after discharge, shared experiences of actively transforming their perspectives and behaviors towards PA. "I want to improve lung function by sticking to morning runs, half an hour of exercise, and cooking." (Inpatient participant #4, male, 63 yrs) "Before falling ill. . . I hardly engaged in any activities, apart from work and commuting. Even household chores were taken care of by my husband. But recently, I enrolled in a swimming class, aiming to become stronger. . ." (Community participant #14, female, 44 yrs)

### Perceived benefits

Inpatient participants described experiences of alleviating fatigue and maintaining physical function through consistent PA before hospitalization. "Since I started cycling regularly, I feel energetic every day, waking up around four in the morning and sleeping around ten at night, with no rest at noon. . ." (Inpatient participant #1, male, 74 yrs) ". . .if I stop exercising for just three days, my limbs become stiff." (Inpatient participant #9, male, 75 yrs)

### Finding enjoyment and confidence

Inpatient participants recalled experiencing joy and gaining confidence and belief in achieving goals through physical activities. "Despite the strenuous nature of hiking. . . my purpose in hiking was to feel the fresh air on the mountain and enjoy the pleasure it brings" (Inpatient participant #8, male, 42 yrs). "Seeing my daily WeChat step count reach 12,000 steps makes me feel

very grounded and accomplished, feeling quite awesome (laughs)" (Inpatient participant #15, male, 68 yrs).

## Passion for physical activity

Inpatient participants expressed a passion for exercise and daily physical activities, continuously adapting and adjusting types and intensities of activities despite facing aging and various health issues. ". . .I used to be a badminton player. . . even after retirement, I played for ten years, one to two hours a day. After falling ill. . . I still take a one-hour walk every day alone" (Inpatient participant #9, male, 75 yrs). "I have a hobby of gardening. . . I can't even remember the names of the flowers anymore, but I still water them" (Inpatient participant #10, male, 96 yrs).

## Cultivating habitual activity

Younger inpatient participants experienced a balance between changing and cultivating activity habits. "Before falling ill, I didn't have the habit of exercising. . . even now, the level of activity is not quite enough. . . I am slowly adapting to change" (Inpatient participant #8, male, 42 yrs). Through consistent PA, many participants gradually formed habits of being active, even during hospitalization. ". . .even during hospitalization, I do breathing exercises every day. . . once the habit is formed, I naturally continue doing it" (Inpatient participant #4, male, 63 yrs).

Offering Participation Opportunities—reiterating the demand for personalized strategies and ideal environmental settings

## Increasing the dose of non-leisure time physical activities

Community residents post-discharge emphasized more efforts in increasing familial, occupational, and commuting physical activities. "To ensure adequate activity, I am responsible for grocery shopping every day." (Community participant #11, female, 67 yrs) "I used to drive to work before, but recently changed jobs. . . I walk for three to five minutes daily for some exercise." (Community participant #14, female, 44 yrs) Additionally, both inpatient and community participants expressed concerns regarding sedentary loads, monotonous and awkward work postures, and prolonged non-restrictive occupational activities exceeding several hours daily. "My job involves logistics management, so there is hardly any activity involved. My family member is a temporary worker in our unit, and she engages in more PA." (Inpatient participant #7, male, 79 yrs) ". . .I resumed [manual labor activities] a week after discharge, working 6 to 7 hours daily. . . My activity levels remained unchanged and did not decrease before and after illness." (Community participant #17, male, 57 yrs)

## Sustaining physical activities during leisure time

Inpatients shared recreational experiences such as hiking, and walking in parks, and medical sports experiences like breathing exercises during hospitalization. "Hiking and staying in the mountains for a while afterward feel much more comfortable. . . Staying in the park longer gives a different feeling compared to bustling urban areas." (Inpatient participant #8, male, 42 yrs) "During hospitalization. . . instructional videos for deep breathing and breathing exercises were played every morning. . . Doing deep breathing and exercises in the morning, I feel my breathing improves. . ." (Inpatient participant #4, male, 63 yrs) However, one elderly participant indicated his negative attitude towards engaging in leisure physical activities and his discomfort with social interactions during participation. "People of our age hardly ever exercise,

some even despise it. This is because, during jogging, the shouting and chanting by some can be quite irritating. . ." (Inpatient participant #7, male, 79 yrs)

## Meeting individualized needs

Some inpatient participants reported that engaging in activities alone helped them avoid the embarrassment caused by declining motor functions due to illness, fully experiencing the enjoyment brought by autonomously choosing PA methods. "Since I walk slower than others, I prefer walking alone so that I don't feel embarrassed." (Inpatient participant #15, male, 68 yrs) "Usually, family and friends ask me to exercise together. . . but I firmly refuse. I prefer doing household chores." (Inpatient participant #6, male, 68 yrs)

## Caring for special population patients

Some participants from special populations, such as oldest-old individuals, widowed people, those living alone, and those living in poverty, have reported facing challenges in engaging in daily activities. "My spouse passed away at the age of 89, and I always water the plants alone. . . My daughter also has to take care of her grandson. . ." (Inpatient participant #10, male, 96 yrs) ". . . I feel short of breath when cooking or bathing, but I still finish because. . . I live alone." (Community participant #12, female, 61 yrs) "Worrying that [activity-induced] exacerbations may lead to returning to the hospital, adding to children's financial and caregiving burdens." (Inpatient participant #3, male, 60 yrs) Additionally, seriously ill and comorbid inpatient participants reported frequent rehospitalizations, severe respiratory distress, and fatigue as reasons for no longer participating in activities. "I spend half of the month in the hospital every month, often feeling fatigued and having difficulty breathing, even walking is a problem. . ." (Inpatient participant #5, male, 66 yrs) "I get very short of breath when active, and I was diagnosed with lung cancer some time ago. . . Now even talking to you feels very difficult (gasping, picking up nasal oxygen tube to inhale)." (Inpatient participant #7, male, 79 yrs)

## Attention to young COPD patients

For young COPD inpatients and community residents, lack of time and companionship affected their enthusiasm for participating in activities. ". . . After work, it's quite tiring, and I just want to go home and lie down. This time tests my willpower [to engage in activities]. . ." (Inpatient participant #8, male, 42 yrs) ". . . I wish a fellow patient was living near me, available at the same time, with whom I could engage in activities, encourage each other, and share experiences about the disease. . . but this threshold is too high! I understand it's hard to come by (sigh)!" (Community participant #14, female, 44 yrs)

## Providing health education resources and reinforcing education

Inpatient participants acknowledged the comprehensive and professional health knowledge provided by healthcare personnel. "During my hospitalization, doctors carefully explained deep breathing and physical activities. Trying them out proved effective, so I persisted." (Inpatient participant #4, male 63 yrs) However, the participants' health conditions during hospitalization affected their access to community health education resources provided by healthcare professionals. ". . .Type II respiratory failure, so no one told me to engage in physical activities." (Community participant #12, female 61 yrs) Some community participants suggested that the effectiveness of education at discharge could be improved. "Upon discharge, the doctor mentioned being more active, but didn't elaborate. I was busy with my discharge, and my focus wasn't on it." (Community participant #13, female 59 yrs)

Participants across different age groups, both in hospitals and communities, described experiences of acquiring knowledge from displays, smartphones, public accounts, and books. "My local amusement parks exhibit 'Brisk walking' and 'Slow walking' signs, encouraging regular visits and memorization." (Inpatient participant #2, male 80 yrs) "I searched for swimming to increase lung capacity on Baidu and TikTok on my phone. . ." (Community participant #14, female 44 yrs) ". . .Books recommend walking more, expanding the chest, which is good for the lungs. . . The hospital also sends me activity knowledge on my phone. . ." (Inpatient participant #15, male 68 yrs)

## Optimizing hospital activity spaces

Inpatient participants described actively engaging in physical activities in the corridors after completing treatment during hospitalization. ". . .I will try to walk in the hallway after the infusion." (Inpatient participant #1, male 74 yrs) However, some participants, due to personal preferences and habits regarding PA, emphasized that the hospital environment restricted their engagement. "This time, because of hospitalization, I can't continue doing household chores. . . There are no conditions for activity in the hospital either. . ." (Inpatient participant #6, male 68 yrs) "I walk at least 1–2 kilometers every morning, [the ward] corridor is too short for proper exercise." (Inpatient participant #9, male 75 yrs)

## Encouraging companionship and support from family and friends

Most inpatient and community participants recalled the physical and psychological benefits of having family, friends, and colleagues accompany and support them during physical activities. "Having a companion allows me to walk longer." (Community participant #13, female 59 yrs) "Walking with friends is enjoyable and uplifting." (Inpatient participant #2, male 80 yrs) Participants discussed reasons for lacking companionship and support, including declining physical function due to illness, family and friends lacking disease and activity knowledge, and support skills, and younger participants wishing to conceal their illness. "My spouse often exercises and invites me along, but I. . . can't keep up with her pace, and friends just invite me to play cards. . ." (Inpatient participant #3, male 60 yrs) ". . .Verbal encouragement from [family] doesn't help much; only family knows about my condition, so there are no friends to support me." (Community participant #14, female 44 yrs)

## Providing accessible and suitable activity spaces and facilities

Inpatient participants shared past experiences of engaging in physical activities in the community, highlighting that weather and seasons directly or indirectly led to reduced or interrupted activities. "I enjoy taking a walk around the park every day, which has hills, bikes, and other activity equipment. However, during the winter season or when it's raining, I refrain from going out as I am susceptible to catching a cold. During these times, I can only be slightly active while walking in the hallway." (Inpatient participant #9, male 75 yrs) "I can only go out when the sun is shining to water the flowers. . ." (Inpatient participant #10, male 96 yrs) Additionally, participants from rural areas expressed varying views on the rural activity environment. "[In the countryside], there's ample space for activities, fresh air, beautiful surroundings, and doing morning exercises is very enjoyable (smiling)." (Inpatient participant #4, male 63 yrs) "We don't have sports fields and facilities in our village. . ." (Inpatient participant #3, male 60 yrs)

Enhancing Participation Capability—addressing strategies for overcoming fears, setting goals, ensuring safety, and adjusting activity levels

## Overcoming fear of physical activity

Inpatient participants perceived engaging in physical activities as challenging both physically and psychologically. "During walks, my feet felt sore, muscles weak, breathing rapid, I felt extremely scared, unable to overcome [the fear of activity] (choked up)." (Inpatient participant #3, male 60 yrs) Confronted with challenges, participants described their ability to overcome fear and maintain positivity. "The notion [of not engaging in activity due to illness-induced fear] is wrong. If you don't breathe smoothly, shallow breathing leads to a vicious cycle. At first, one or two times might feel uncomfortable, but after getting used to it several times, breathing naturally becomes smooth." (Inpatient participant #4, male 63 yrs)

## Setting goals and executing plans

In many instances, participants continued to actively set activity goals and diligently execute plans. "I walk for 40 minutes every morning, 40–60 minutes in the evening, and I must achieve 10,000–12,000 steps on WeChat every day. If it's windy or raining, I walk around my doorstep to achieve 5,000–6,000 steps." (Inpatient participant #15, male 68 yrs) However, as resistance to plan execution increased, community participants began seeking ways to reconcile activities. "I feel easily fatigued, so I need longer sleep to ensure normal daytime activities." (Community participant #14, female 44 yrs)

## Self-Activity safety management

Young inpatient participants, upon mastering activity skills, began showing concerns about activity safety. "With increasing age and declining physical function, sudden conditions may occur. It's best to have family, companions, or medical personnel accompany and assist." (Inpatient participant #8, male 42 yrs) An elderly participant described his experience of falling during activity. "I have over a hundred types of flowers at home, needing watering every day, but I can't carry the water basin, trim, and loosen the soil now. Once, I fell while carrying water... All I can do is water the flowers (sigh)." (Inpatient participant #10, male 96 yrs)

## Assessing and adjusting activity dosage

Inpatient and community participants discussed their experiences of assessing and adjusting activity types, intensity, frequency, and duration based on physical condition or time. "[Activity intensity] can't be compared with dancing and brisk walking friends. Usually, when I want to walk at my own pace, I'll walk alone or do chest expansion exercises." (Community participant #11, female 67 yrs) "The onset of the disease is time-limited, and the habit of being active can be developed by increasing the number of activities during the stabilization period." (Inpatient participant #15, male 68 yrs) Some participants also described how they assessed and adjusted the dosage of their activities during the activity itself, prioritizing appropriate activity based on their ability, taking appropriate rest, and progressing gradually. "Every time I go up or down a slope, I assess my own strength and breathing level, then gradually ascend or descend, taking breaks in between." (Inpatient participant #9, male 75 yrs) "When I sweat during activity, I stop to rest, then slow down my walking pace." (Inpatient participant #16, male 64 yrs) Additionally, community participants reported choosing to interrupt activity when the dosage couldn't be adjusted during activity. "... I'll take more breaks, drink water, or just stop." (Community participant #17, male 57 yrs)

## Discussion

This qualitative study aims to comprehensively understand the motivational, opportunistic, and capability aspects influencing PA engagement among COPD patients. Seventeen participants, either hospitalized or from the community, described the intrinsic reasons and immediate drives for participating in physical activities. Additionally, the effective components within the external objective environment that facilitated their engagement in activities were reported, along with the PA skills required or already mastered.

The impact of motivation on PA engagement among COPD patients is well-documented [17, 32–34]. Known influencing factors on motivation encompasses the concept of PA, the quality of motivation, available opportunities, perceived benefits, convenient schemes, retirement, social support, seasons, physical condition, family responsibilities, transportation, physical capabilities, emotional enhancements, fear of breathlessness, and comorbidities. However, our study indicates that motivational factors, including understanding the significance, desire to alleviate symptoms, deriving pleasure and confidence, passion for PA, and cultivating activity habits, also exert specific influences on participation behavior. Particularly, due to the desire to ameliorate symptoms caused by the illness, some participants actively transition their intentions and behaviors regarding PA after discharge. Additionally, we provided a detailed account of the opportunities and capabilities influencing motivational factors. Inpatients tended to share experiences of engaging in recreational and therapeutic physical activities during their leisure time, while community participants focused more on familial, occupational, and commuting physical activities. Moreover, inpatients presented with more personalized needs, faced limitations in both inpatient and community activity settings and encountered safety hazards. Importantly, some critically ill inpatients with comorbidities ceased participation in activities, further corroborating findings from previous studies [35, 36].

Focusing on the motivational, opportunistic, and capability factors noted in our research and prior studies about COPD patients' PA engagement, interventions should be prioritized aimed at improving related areas. These might include: inducing healthcare institutions' attention and increasing funding for PA programs; amplifying interdisciplinary training among healthcare practitioners to enhance the potency of activity guidance and counseling; and addressing personalized patient necessities to foster an increment in non-leisure time PA levels.

Prioritizing the role of healthcare institutions in recognizing and funding physical activities could yield maximal benefits for COPD patients. Participant narratives regarding engagement in physical activities reflect a desire for respiratory improvement and enhanced physical fitness through activities. However, concerns about the medical-economic burden and caregiving responsibilities stemming from activities serve as deterrents to sustained participation. Simultaneously, participants express a need for accessible activity locations and facilities to mitigate demotivation caused by weather conditions like summer heat or rainy days and physical environmental constraints like rural settings or hospital environments. Notably, while GOLD guidelines advocate for reducing disease progression through PA, investments by healthcare institutions in promoting PA remain scarce. A study [37] indicates potential cost savings of £35 per month and cost-effectiveness of £202 per month through PA in the UK COPD population. Encouraging exercise solely through medical advice could result in healthcare cost savings, and if PA is subsidized, it could still prove cost-effective. Hence, future researchers should develop and apply substantial evidence supporting the role of PA, and garner attention and funding from healthcare institutions, thus promoting and sustaining PA as an effective method for encouraging COPD patient participation.

Enhancing the training and collaboration among multidisciplinary personnel such as physicians, nurses, and physiotherapists may also aid in promoting COPD patient engagement in

PA. Our study corroborates previous findings that understanding the significance is a crucial motivator for COPD patients' PA involvement [32]. However, the reality is that guidance provided by healthcare professionals exhibits variability in its impact on participants' activity, with some expressing never receiving relevant advice [23]. Patients perceive general practitioners' focus on medication and smoking cessation rather than PA and diet. Due to a lack of knowledge regarding PA tolerance levels, patients avoid activities that elevate heart rates [38]. Healthcare professionals acknowledge the importance of PA for COPD patients and the patients' low levels of PA. However, specific discussions are infrequent, despite knowledge of PA guidelines. Due to constraints like time, treatment priorities, and a lack of expertise, they often prefer physiotherapists to provide more comprehensive assessments and recommendations [39]. Extensive research demonstrates the significant roles of primary care physicians [40], nurses [41], and occupational therapists [42] in providing respiratory care services, setting goals, and delivering self-management education, particularly through the provision of remote healthcare services based on multidisciplinary teamwork, which prioritizes the needs of various stakeholders [43]. Additionally, patients find remote media training interventions meaningful, aiding in their adherence to basic activities in daily life [44]. Beneficial approaches may also encompass health coaching (e.g., goal-setting, motivational interviews, and health education) [45] and personalized PA consultations [46]. Future research should further explore communication and collaboration models within multidisciplinary teams to expand effective and inclusive intervention approaches and methods, thereby enhancing the guidance efficacy for COPD patient PA participation.

Meeting individualized patient needs to promote an increase in non-leisure time PA levels may also encourage them to maintain PA in the long term. In our study, elderly, widowed, impoverished, critically ill, comorbid, isolated, and young participants also described the challenges of participating in activities, which may need to be addressed by increasing the dosage of Vigorous Intermittent Lifestyle Physical Activity (VILPA) [47], while also preventing Physical Activity-Related Injuries (PARI) [48]. Two large-sample studies by Stamatakis and his team [49, 50] confirmed that minimal VILPA is associated with a lower risk of cancer and significantly reduced all-cause, cardiovascular, and cancer mortality. It is worth noting that it may also rapidly improve cardiorespiratory health [51]. However, its relevance to COPD should be further investigated. Daily 4–5 minutes of VILPA may be a promising intervention for promoting health in COPD patients who are unable or unmotivated to exercise during leisure time. Additionally, community-based personalized outdoor walking interventions [52], empowering the use of electronic health tools for a sense of control [53], and increasing the dosage of Light Physical Activity (LPA) [54] are crucial for maintaining or enhancing the PA levels of COPD patients.

Furthermore, measures suggested to meet the individualized needs of COPD patients include regular assessment of the degree of respiratory difficulty, balance control ability, and osteoporosis status [55–57]; sharing activity data with participants using activity trackers [58, 59]; utilizing new technologies for systematic and long-term risk monitoring [60]; encouraging female and comorbid patients to participate in PA programs [61, 62]; providing online peer support or digital social interventions for younger participants [63]; and paying attention to the acceptance of patient role behavior in Asian culture, along with providing psychological care [64].

Finally, as an open and inclusive behavioral change theory tool, the MOA model can flexibly translate each core concept into specific actionable concepts when studying the participation and maintenance of PA in specific COPD patients. This approach facilitates the identification of specific reasons for specific behaviors, serving both implementation purposes and remaining accountable to scientific scrutiny and revision, thus providing information for

simple and effective intervention measures to deepen the understanding of the drivers of PA participation and maintenance behaviors and to inform the future. Therefore, further development and refinement of this theory and intervention measures hold considerable promise.

Purposeful sampling contributes to incorporating experiences covering a broad spectrum of information, including those of elderly, young, severely ill, and comorbid participants. Nevertheless, this study still faces certain limitations. Firstly, our sample is confined to two wards in one hospital, with participants exclusively of Asian descent, thus limiting the generalizability of the research findings. Additionally, over the past six months, only five discharged participants have been recruited via telephone, including all four female participants in this study. The lack of non-verbal communication might hinder the establishment of intimacy and trust during face-to-face interviews, potentially impacting their responses or the information provided. Secondly, our analysis employed the rigorous Colaizzi seven-step data analysis method; however, alternative interpretations are possible. Furthermore, motivation, a fundamental concept in psychology within the MOA model, cannot be directly observed. Drawing conclusions based on stimulus situations and behavioral responses may involve subjectivity, and the model might overlook influencing factors such as social and cultural backgrounds, personal beliefs, and values.

## Conclusions

This study has provided us with a deeper understanding of the motivations, opportunities, and capabilities for engaging in physical activities from the perspective of COPD patients. It lays the groundwork for developing personalized interventions based on interdisciplinary teams. Delving into the motivational aspects of patient engagement in physical activities and focusing on providing opportunities for participation, enhancing engagement capabilities emerge as crucial priorities. Seeking recognition and funding from healthcare institutions, strengthening training and collaboration among multidisciplinary teams, and meeting individualized needs and preferences to promote an increase in non-leisure time PA levels should be considered.

## Supporting information

**S1 Appendix. COREQ: 32-item checklist.**
(DOCX)

**S2 Appendix. Interview guide.**
(DOCX)

## Acknowledgments

We express our gratitude to all participants who contributed their time and expertise to this study. Additionally, we extend our thanks to the medical and nursing staff of the Respiratory Medicine Department at Tongji Medical College, Huazhong University of Science and Technology, affiliated with Union Hospital, for facilitating the recruitment of participants.

## Author Contributions

**Conceptualization:** Yuanyu Liao, Jiaohua Yu, Yuxin Zhan, Yunfang Liu.

**Data curation:** Yuanyu Liao.

**Formal analysis:** Yuanyu Liao.

**Funding acquisition:** Jiaohua Yu.

**Investigation:** Yuanyu Liao, Yaoling Zhou.

**Methodology:** Yuanyu Liao, Jiaohua Yu, Yuxin Zhan, Yunfang Liu, Huan Wang, Xinghong Liu, Weiwei Wang, Yu Ma, Fenfen Lan.

**Project administration:** Jiaohua Yu.

**Resources:** Jiaohua Yu.

**Supervision:** Jiaohua Yu.

**Visualization:** Yuanyu Liao.

**Writing – original draft:** Yuanyu Liao, Yaoling Zhou.

**Writing – review & editing:** Jiaohua Yu, Yuxin Zhan, Yunfang Liu, Huan Wang, Xinghong Liu, Weiwei Wang, Yu Ma, Fenfen Lan.

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
