## [Decision Letter · Decision Letter 0]

15 Feb 2024

PONE-D-24-00066Motivating Factors for Physical Activity Participation among Individuals with Chronic Obstructive Pulmonary Disease: A Qualitative Study Applying the Motivation, Opportunity, and Ability ModelPLOS ONE

Dear Dr. Yu,

Thank you for submitting your manuscript to PLOS ONE. After careful consideration, we feel that it has merit but does not fully meet PLOS ONE’s publication criteria as it currently stands. Therefore, we invite you to submit a revised version of the manuscript that addresses the points raised during the review process.

We look forward to receiving your revised manuscript.

Kind regards,

Petra Czarniak, PhD

Academic Editor

PLOS ONE

Reviewers' comments:

Reviewer's Responses to Questions

**Comments to the Author**

1. Is the manuscript technically sound, and do the data support the conclusions?

Reviewer #1: Yes

Reviewer #2: Yes

2. Has the statistical analysis been performed appropriately and rigorously? 

Reviewer #1: Yes

Reviewer #2: I Don't Know

3. Have the authors made all data underlying the findings in their manuscript fully available?

Reviewer #1: Yes

Reviewer #2: Yes

4. Is the manuscript presented in an intelligible fashion and written in standard English?

Reviewer #1: Yes

Reviewer #2: Yes

5. Review Comments to the Author

Reviewer #1: This is an interesting, informative and well-conducted piece of research which has found indication that inspiring participation motivation, providing participation opportunities, and enhancing participation capability are key strategies for promoting effective engagement in physical activity among individuals with COPD. The findings are of clinical relevance and have the potential to be translated into practice globally.

Reviewer #2: Thank you for the opportunity to review the manuscript focusing on exploring the motivational factors for physical activity engagement for people with COPD, who are either hospitalised or discharged within the last 6 months. Seventeen adults were interviewed and the findings framed according to the Motivation-Opportunity-Ability (MOA) model. The manuscript is well written and provides insightful findings on some of the salient barriers and potential enablers for people with COPD initiating and maintaining physical activity. Although the authors give some justification for adopting the MOA model, there is little appraisal of the framework in-light of other established models in the Introduction and scant reflection (acknowledgement of limitation) in the Discussion. What makes the MOA model more suitable than say, the COM-B model or transtheoretical model? Such considerations are worth reviewing no revision. My specific comments can be found below:

Abstract:

Well-structured, but the Methods does not mention that some participants were 'discharged' and not hospitalised, as stated in the manuscript (full) Methods.

Introduction:

There is a vast literature on physical activity maintenance, including for people living with long-term conditions, which is not mentioned directly in the Introduction. A brief overview of PA maintenance is needed, as it is directly relevant to COPD and future interventions incorporating behaviour change (e.g. MOA).

MOA model - greater justification is required for selecting this particular model, at least highlighting succinctly other existing frameworks that could also be used to inform future PA interventions.

Materials and Methods:

The Study Design (2.2) subsection should come before 2.1 Research Team and Reflexivity.

Page 4, lines 3-4: "...the researcher established trustful relationship with the participants through interactions with responsible nurses." How exactly did the researcher establish relationships with participants? Particularly when the interactions appear limited to nursing staff.

Subsection 2.2.1. This heading is inappropriate, please revise as separate.

It is suggested that face-to-face interviews were conducted with hospitalised patients, as well as those discharged in the last 6-months. This requires more focus and appraisal throughout the manuscript, particularly in the Results (either showing demographics of respective groups, and/or highlighting perspectives in the quotes (by providing brief demographics with the p#). This would help the reader's understanding and interpretation of the findings greatly. It should also be reflected upon in the Discussion, potentially as a limitation.

2.2.2 Data Collection

References are required throughout this subsection to support your approaches, notably in assessing data saturation.

2.3 Data Analysis

Although the following paragraph highlights partial roles, I recommend that on revision you provide author initials to highlight what researchers were involved at specific steps of the Colaizzi Method.

Results

The results presents arguably too many quotes and too little descriptive narrative from the authors. Please consider revising for balance throughout the Results.

It is important for the reader to know what quotes came from the hospitalised, and form the discharged, community dwelling. Furthermore, frame quotes with relevant and concise demographics (e.g. Participant #17, male 45 yrs)

Page 7, final line: "...and I don't feel short of breath." Participant #4 and #9 are stated. It is unclear who said what. Please check other similar occurences in the Results.

Discussion

Focused and aligned to the aims, and research findings. The Discussion would benefit from appraisal of the literature on the maintenance of physical activity and linking the MOA model to behaviour change theory (perhaps linking to future work). Also, please given some discussion to whether there were differences in the experiences of participants who were hospitalised and discharged/community dwelling. I would suspect the severity of COPD symptoms (and/or physical health) to be greater (and worse) for those hospitalised, as well as fewer opportunities to engage in PA, and therefore accounts would be expected to be unique between sub-samples.

Limitation - any reflections on the MOA Model? At least comment on whether MOA did not suit any observations from participants

6. PLOS authors have the option to publish the peer review history of their article (what does this mean?). If published, this will include your full peer review and any attached files.

Reviewer #1: **Yes: **Dr Sarah Markham

Reviewer #2: **Yes: **James P. Gavin

---

## [Author Response · Author response to Decision Letter 0]

20 Feb 2024

Dear Dr. Petra Czarniak,

We would like to express our gratitude to you and the reviewers, Dr. Sarah Markham and James P. Gavin, for providing valuable feedback on our manuscript titled "Motivating Factors for Physical Activity Participation among Individuals with Chronic Obstructive Pulmonary Disease: A Qualitative Study Applying the Motivation, Opportunity, and Ability Model." We appreciate the time and effort invested in evaluating our work and offering constructive suggestions for improvement. We have carefully considered each comment and have made the necessary revisions to address the concerns raised. Below, we provide a detailed response to each reviewer's comments:

Reviewer: PLOS ONE

[Reviewer's Comment 1] 

Response: We have diligently adhered to the style templates provided by the editors to ensure that the manuscripts meet PLOS ONE's style and file naming requirements. Additionally, the table files have been uploaded to the Pre-submission Analysis and Conversion Engine (PACE) digital diagnostic tool, aligning with PLOS's specifications. Please refer to Table 1 for details.

[Reviewer's Comment 2] 

Response: The corresponding author of this article has provided and validated the ORCID iD in Editorial Manager. The ORCID iD is 0009-0008-9002-5370.

Reviewer #1: Dr. Sarah Markham

[Reviewer's Comment]

Response: We sincerely appreciate your thoughtful and encouraging assessment of our manuscript titled "Motivating Factors for Physical Activity Participation among Individuals with Chronic Obstructive Pulmonary Disease: A Qualitative Study Applying the Motivation, Opportunity, and Ability Model." Your acknowledgment of the study as an interesting, well-executed, and clinically significant research endeavor is truly gratifying.

We are thrilled that you recognize the potential global impact of our findings and their applicability to real-world clinical practices. Your positive evaluation serves as motivation to continue our efforts in contributing valuable insights to the field of chronic obstructive pulmonary disease management.

We are grateful for the absence of specific revision suggestions, indicating that our manuscript aligns well with the standards of PLOS ONE. Your feedback encourages us to maintain the rigor and relevance of our research, and we are committed to ensuring the clarity and coherence of our work.

Once again, we express our gratitude for your positive appraisal and your time devoted to reviewing our manuscript. We look forward to the possibility of contributing to the dissemination of knowledge in PLOS ONE.

Reviewer #2: James P. Gavin

[Reviewer's Comment 1: Description of participants in the Methods section of the abstract should include both inpatient and outpatient individuals]

Response: We appreciate Dr. Gavin's feedback on the need for a detailed description of participants from different sources in the Methods section of the abstract. In response to this comment, we have supplemented this section to align with the overall manuscript. Please refer to the revised manuscript on page 2, lines 28-29.

[Reviewer's Comment 2: Introduction should address the maintenance of physical activity]

Response: Regarding the suggestion to incorporate references in the introduction regarding the maintenance of physical activity, including in long-term patients, we have carefully revised the introduction to address this issue. Please see the revised manuscript on page 4, line 82, and page 5, lines 83-86. These modifications enhance the overall coherence and readability of the manuscript.

[Reviewer's Comment 3: The introduction should evaluate the reasons for choosing the MOA model from the perspective of other existing models]

Response: We appreciate Dr. Gavin's insightful suggestions for reexamining and revising other existing models and the MOA model. In response to this suggestion, we have added paragraphs to the introduction section in the revised manuscript to address this issue. Please see the revised manuscript on page 5, lines 95-105, and page 6, lines 106-116. These changes improve the clarity and rigor of our study.

[Reviewer's Comment 4: Adjustments in the Materials and Methods section regarding paragraph order and content addition]

Response: The subsection "Study Design 2.2" has been adjusted to precede "2.1 Research Team and Reflexivity." Regarding the concern about how researchers establish relationships with participants, especially when interactions seem limited to healthcare providers, detailed explanations have been provided in the revised manuscript on page 9, lines 162-172, and page 10, lines 173-174. The title of subsection 2.2.1 has been modified to "Participant Selection and Recruitment." Percentage data for inpatient and within 6 months post-discharge participants have been added to Table 1 on page 8. Two references supporting methods for assessing data saturation have been included in the data collection subsection on page 9, lines 156-157. The data analysis section in the revised manuscript on page 10, lines 176-190, now includes author initials and highlights researchers' involvement in specific steps of the Colaizzi method. We believe these revisions strengthen the validity and reliability of our study results.

[Reviewer's Comment 5: Balance between citations and narrative in the Results section, differentiate between inpatient and post-discharge community residents]

Response: We have carefully reviewed our Results section and made necessary adjustments, including reducing citations, increasing narrative content for a balanced presentation, and explicitly noting which citations are from inpatient versus post-discharge community residents. Relevant concise demographic data (e.g., Participant 17, male, 45 years old) have been added to citations. We have also examined and corrected descriptions that simultaneously state quotes from two participants. Please refer to the revised manuscript on pages 11-19.

[Reviewer's Comment 6: Add content and reflection to the Discussion and Limitations sections]

Response: In the revised Discussion section, we have added content on maintaining physical exercise (page 22, lines 453-455), connecting the MOA model with behavior change theories and future work (page 23, lines 463-469), and highlighting differences in experiences between inpatient and post-discharge/community-residing participants (page 20, lines 393-403). In the revised Limitations section, we have included reflections and comments on the MOA model (page 23, lines 478-481) and limitations related to the study population (page 23, lines 472-476).

In conclusion, we believe that the revisions made in response to the reviewers' comments have significantly improved the quality and clarity of our manuscript. We are grateful for the opportunity to address these valuable suggestions and are confident that the revised manuscript meets the standards of PLOS ONE. We appreciate your consideration of our work and look forward to the opportunity for publication.

Thank you once again for your time and consideration.

Best regards,

Yuanyu Liao

Master's research student

e-mail: Katrina9820@126.com

---

## [Decision Letter · Decision Letter 1]

26 Mar 2024

PONE-D-24-00066R1Motivating factors for physical activity participation among individuals with chronic obstructive pulmonary disease: a qualitative study applying the motivation, opportunity, and ability modelPLOS ONE

Dear Dr. Yu,

 Thank you for submitting your manuscript to PLOS ONE. After careful consideration, we feel that it has merit but does not fully meet PLOS ONE’s publication criteria as it currently stands. Therefore, we invite you to submit a revised version of the manuscript that addresses the points raised during the review process.

We look forward to receiving your revised manuscript.

Kind regards,

Petra Czarniak, PhD

Academic Editor

PLOS ONE

Journal Requirements:

**Additional Editor Comments:**

Thank you for submitting your revised manuscript to PLOS ONE. It is very much appreciated that the authors have responded in a focused manner to the reviewers’ comments. However, there remain a few superficial errors in grammar and text, relating to the new added text. For example, on revision of the manuscript, a number of the author names are written entirely in capital letters. Please check the manuscript for typographical errors, formatting and grammar. If the authors proof-read and amend this version, the manuscript should be worthy of publication. Therefore, we invite you to submit a revised version of the manuscript that addresses these points.

Reviewers' comments:

Reviewer's Responses to Questions

**Comments to the Author**

1. If the authors have adequately addressed your comments raised in a previous round of review and you feel that this manuscript is now acceptable for publication, you may indicate that here to bypass the “Comments to the Author” section, enter your conflict of interest statement in the “Confidential to Editor” section, and submit your "Accept" recommendation.

Reviewer #1: All comments have been addressed

Reviewer #2: (No Response)

2. Is the manuscript technically sound, and do the data support the conclusions?

Reviewer #1: Yes

Reviewer #2: (No Response)

3. Has the statistical analysis been performed appropriately and rigorously? 

Reviewer #1: Yes

Reviewer #2: Yes

4. Have the authors made all data underlying the findings in their manuscript fully available?

Reviewer #1: Yes

Reviewer #2: Yes

5. Is the manuscript presented in an intelligible fashion and written in standard English?

Reviewer #1: Yes

Reviewer #2: Yes

6. Review Comments to the Author

Reviewer #1: I have reviewed the revised manuscript, the reviewer comments and the author's responses and am happy with the revisions made.

Reviewer #2: Dear authors,

Thank you for the opportunity to review this revision of the submitted manuscript, I feel that on revision you have responded to the comments of the reviewers adequately, particularly in terms improving the Introduction and rationale for the theoretical model used.

However, there remain a few superficial errors in grammar and text, relating to the new added text. For example, a number of the author names added on revision to the Introduction, include those in upper-case and there are lapses in grammar/clarity. Please ensure that the manuscript is proofed and check again for reference formatting.

Yours sincerely,

Reviewer 2

7. PLOS authors have the option to publish the peer review history of their article (what does this mean?). If published, this will include your full peer review and any attached files.

Reviewer #1: No

Reviewer #2: **Yes: **James Gavin

---

## [Author Response · Author response to Decision Letter 1]

29 Mar 2024

Dear Editor and Reviewers,

We would like to express our gratitude for your insightful comments and suggestions on our manuscript, "Motivating factors for physical activity participation among individuals with chronic obstructive pulmonary disease: a qualitative study applying the motivation, opportunity, and ability model." We have carefully considered each point and made the following revisions:

1. Acknowledgment of Funding: We have added a second grant information: the Hubei Provincial Department of Finance 2024 Research Program (08.01.24011) is also held by JY, the corresponding author of this paper, and a detailed financial disclosure statement was made in the cover letter.

2. Language and Formatting: We undertook a thorough review and revision of grammar, presentation, and formatting throughout the manuscript, particularly new additions to improve clarity and correctness. This included further revisions to the abstract; harmonizing the abbreviations for "chronic obstructive pulmonary disease," "physical activity," and "motivation-opportunity-ability;" and ensuring consistency in the formatting of the authors' names throughout the text, including the added theoretical description paragraph in the Introduction section and the researcher's initials in the Methods section. In addition, we refined the concluding sentences in the Discussion section.

3. Table Revision: We have revised "Occupation" to "Occupational Status" in Table 1, adjusted the legends to meet PLOS ONE formatting requirements, and re-uploaded them to the PACE digital diagnostic tool to ensure compliance.

4. Consistency in Themes: We have ensured consistency between the themes presented in the Results section and the descriptions provided in the abstract.

5. Reference formatting: All references were reviewed and corrected to comply with PLOS ONE guidelines, including formatting adjustments such as replacing commas with semicolons after the year of publication, deleting specific months and days, and deleting redundant spaces. We also completed missing page numbers for references 2, 3, 5, and 16; reconfirmed the absence of doi numbers in article 6, providing PMIDs only; added author names for references 8 and 9, made appropriate adjustments to reference 12, abbreviated journal names for reference 25, and ensured that references 1, 5, 10-17, 21, 24-27, 29, 32, 34, 36, 40, 42, 44-46, 48, 50, 52, 54, 57-58, 60-61 were adjusted to lower case except for initial letters.

6. Update of Supplementary Material: We have updated Appendix S1 (the Consolidated Criteria for Reporting Qualitative Research [COREQ]) to reflect all corresponding modifications.

7.Protocols.io Submission: Since we have provided a comprehensive research methodology in our paper, we confirm that it does not apply to submissions to protocols.io.

Once again, we sincerely appreciate the time and effort invested in reviewing our paper. We believe that the revisions we have made address all the concerns raised by the editor and reviewers, and we are confident that these revisions have significantly improved the quality and clarity of our manuscript.

Thank you for your thorough assessment, and we kindly request that you consider our revised manuscript for acceptance into PLOS ONE. We remain at your disposal for any further clarifications or modifications if necessary.

Sincerely, 

Yuanyu Liao

First Author

---

## [Editor Report · Decision Letter 2]

9 Apr 2024

PONE-D-24-00066R2Motivating factors for physical activity participation among individuals with chronic obstructive pulmonary disease: a qualitative study applying the motivation, opportunity, and ability modelPLOS ONE

Dear Dr. Yu,

Thank you for submitting your revised manuscript to PLOS ONE. In reviewing the manuscript, it does not fully align with the document that shows the tracked changes. Further, some adjustments to grammar are recommended and all references should be cited in the manuscript. For example:

Line 83 Please cite the study by Dragnish et al.

Line 265/66 Please check grammar of the sentence ‘However, a raised barriers faced in participating in public sports, especially employee and senior citizen sports.’ In its current form the sentence does not make sense.

Line 267/268 Pease check grammar of the sentence ‘People our age hardly exercise, some even dislike it because some people like to shout or chant slogans while which irritates me.’ Consider removing the word ‘while’ from the sentence or clarify if this should be ‘while running’.

Line 307/308 Do the authors mean ‘Near my home, there are signs in the park…..’

These are only some example.

We look forward to receiving your revised manuscript.

Kind regards,

Petra Czarniak, PhD

Academic Editor

PLOS ONE
---

## [Author Response · Author response to Decision Letter 2]

11 Apr 2024

Subject: Reply to Reviewer's Comment for Article ID PONE-D-24-00066R2

Dear Dr. Petra,

We would like to express our gratitude for your insightful comments and suggestions on our manuscript entitled "Motivating factors for physical activity participation among individuals with chronic obstructive pulmonary disease: a qualitative study applying the motivation, opportunity, and ability model" submitted to PLOS ONE. We appreciate the opportunity to address your concerns and improve the quality of our research. Below, we provide a detailed response to each of your points:

Discrepancy between Manuscript and Revised Manuscript with Track Changes: We sincerely apologize for any inconsistencies between the Manuscript and the Revised Manuscript with Track Changes. Upon thorough review, we discovered that the discrepancies arose from the automatic formatting changes that occurred when our Word document was uploaded to the system and converted into a PDF file. For instance, in line 83 of our original Word document, we referenced the study by Dragnich et al., which unfortunately did not appear as [11] after uploading. To mitigate such issues in the future, we have diligently cross-checked both the manuscript in Word format and the system-generated PDF version to ensure that all references are properly cited and that there are no discrepancies between the manuscript and the tracked changes document.

Grammar Adjustments throughout the Manuscript: In response to your suggestions, we have made several grammatical adjustments throughout the manuscript to enhance clarity and accuracy. Specifically, we have rewritten and polished lines 265/266 and 372/373 to better articulate our analysis of the interview data. Additionally, we have reviewed and revised lines 267/268 to maintain logical consistency and prevent grammatical errors. We have also refined lines 277/278 to accurately convey the concept of "oldest-old individuals." Furthermore, we have polished lines 307/308, 315/316, 334/336, and 370/371 to provide a clearer expression of the interviewees' statements.

We believe that these revisions have strengthened the overall quality and academic rigor of our manuscript. Thank you once again for your invaluable feedback and guidance throughout this process. We are confident that the improvements made will contribute to the scholarly excellence of our work.

Sincerely,

Yuanyu Liao

First Author

e-mail: Katrina9820@126.com

---

## [Editor Report · Decision Letter 3]

25 Apr 2024

PONE-D-24-00066R3Motivating factors for physical activity participation among individuals with chronic obstructive pulmonary disease: a qualitative study applying the motivation, opportunity, and ability modelPLOS ONE

Dear Dr. Yu,

Thank you for submitting your revised manuscript to PLOS ONE. After careful consideration, we feel that it has merit but does not fully meet PLOS ONE’s publication criteria as it currently stands. Currently, the COREQ checklist does not align with the manuscript. Therefore, we invite you to submit a revised version of the COREQ checklist, as part of the manuscript.

Please submit your revised manuscript COREQ checklist by Jun 09 2024 11:59PM. If you will need more time than this to complete your revisions, please reply to this message or contact the journal office at plosone@plos.org. Please include the following items when submitting your revised manuscript:A rebuttal letter that responds to each point raised by the academic editor and reviewer(s). You should upload this letter as a separate file labeled 'Response to Reviewers'.A marked-up copy of your manuscript that highlights changes made to the original version. You should upload this as a separate file labeled 'Revised Manuscript with Track Changes'.An unmarked version of your revised paper without tracked changes. You should upload this as a separate file labeled 'Manuscript'.If applicable, we recommend that you deposit your laboratory protocols in protocols.io to enhance the reproducibility of your results. Protocols.io assigns your protocol its own identifier (DOI) so that it can be cited independently in the future. For instructions see: https://journals.plos.org/plosone/s/submission-guidelines#loc-laboratory-protocols. Additionally, PLOS ONE offers an option for publishing peer-reviewed Lab Protocol articles, which describe protocols hosted on protocols.io. Read more information on sharing protocols at https://plos.org/protocols?utm_medium=editorial-email&utm_source=authorletters&utm_campaign=protocols.

We look forward to receiving your revised manuscript.

Kind regards,

Petra Czarniak, PhD

Academic Editor

PLOS ONE
---

## [Author Response · Author response to Decision Letter 3]

27 Apr 2024

Dear Dr. Petra,

Thank you for your thoughtful review on our manuscript "Motivating factors for physical activity participation among individuals with chronic obstructive pulmonary disease: a qualitative study applying the motivation, opportunity, and ability model," submitted to PLOS ONE (Manuscript ID: PONE-D-24-00066R3). We've addressed your concerns and made the following revisions:

We've updated the COREQ checklist to align with the manuscript. The revised COREQ checklist is attached with tracked changes for clarity.

We've corrected the missing letters in the 9th reference and ensured all references are accurate and meet PLOS ONE standards.

We've tried several times to fix line 191 on page 10 but haven't found a solution yet for the formatting inconsistency post-upload.

Thanks again for your detailed review and helpful suggestions. We trust these revisions improve our manuscript. Please let us know if further adjustments are needed.

Best regards,

Yuanyu Liao

First Author

e-mail: Katrina9820@126.com

---

## [Editor Report · Decision Letter 4]

2 May 2024

Motivating factors for physical activity participation among individuals with chronic obstructive pulmonary disease: a qualitative study applying the motivation, opportunity, and ability model

PONE-D-24-00066R4

Dear Dr. Yu,

We’re pleased to inform you that your manuscript has been judged scientifically suitable for publication and will be formally accepted for publication once it meets all outstanding technical requirements.

Kind regards,

Petra Czarniak, PhD

Academic Editor

PLOS ONE
---

## [Editor Report · Acceptance letter]

14 May 2024

PONE-D-24-00066R4 

PLOS ONE

Dear Dr. Yu, 

I'm pleased to inform you that your manuscript has been deemed suitable for publication in PLOS ONE. Congratulations! Your manuscript is now being handed over to our production team.

Kind regards, 

on behalf of

Dr. Petra Czarniak 

Academic Editor

PLOS ONE